# Characterization of Bioactive Colored Materials Produced from Bacterial Cellulose and Bacterial Pigments

**DOI:** 10.3390/ma15062069

**Published:** 2022-03-11

**Authors:** Lúcia F. A. Amorim, Raul Fangueiro, Isabel C. Gouveia

**Affiliations:** 1FibEnTech Research Unit, Faculty of Engineering, University of Beira Interior, 6200-001 Covilhã, Portugal; lucia.amorim@ubi.pt; 2Centre for Textile Science and Technology (2C2T), University of Minho, 4800-058 Guimarães, Portugal; rfangueiro@dem.uminho.pt

**Keywords:** bacterial cellulose, bacterial pigments, antimicrobial, antioxidant, prodigiosin, flexirubin

## Abstract

A Bacterial Cellulose (BC) film was developed and characterized as a potential functional bioactive material. BC films, obtained from a microbial consortium of bacteria and yeast species, were functionalized with the bacterial pigment prodigiosin, produced by *Serratia plymuthica*, and flexirubin-type pigment, from *Chryseobacterium shigense*, which exhibit a wide range of biological properties. BC was successfully functionalized at 15% over the weight of the fiber at 40 °C during 60 min, and a color strength of 1.00 ± 0.01 was obtained for BC_prodigiosin and 0.38 ± 0.02 for BC_flexirubin-type pigment. Moreover, the BC films showed moderate hydrophilic character following alkaline treatment, which was maintained after both pigments were incorporated. The porosity and mechanical performance of the functionalized BC samples also remained unaffected. Furthermore, the BC samples functionalized with prodigiosin presented antibacterial activity and were able to inhibit the growth of pathogenic bacteria *Staphylococcus aureus* and *Pseudomonas aeruginosa*, with inhibition rates of 97.89 ± 0.60% and 85.12 ± 0.17%, respectively, while BC samples functionalized with flexirubin-type pigment exhibited the highest antioxidant activity, at 38.96 ± 0.49%. This research provides an eco-friendly approach to grant BC film-based material with color and advantageous bioactive properties, which can find application in several fields, especially for medical purposes.

## 1. Introduction

Bacterial Cellulose (BC) is a remarkable biomaterial produced by microorganisms, exhibiting high mechanical strength, due to its web-like network structure, high crystallinity, high water holding capacity, high porosity, and high degree of polymerization [1,2,3] which makes it a very useful biomaterial in many different fields [1].

The BC applicability ranges from the pharmaceutical industry, cosmetics, biomedicine, paper additives, electronics, and textile industry, to a food substitute, food additives, food packaging, and filtration membranes [4,5]. Despite all the superior properties aforementioned, for most of these applications, BC is often endowed with bioactive properties, such as antimicrobial and antioxidant ability, in order to surpass any restrictions to its use. For example, in the development of wound dressing materials for medical applications, bioactive properties have been reported to significantly reduce the complications caused by bacterial colonization of wounds, contributing to an accelerated healing process [6].

Furthermore, agents often used to provide antimicrobial properties, such as metallic nanoparticles, have limited applicability due to fabrication through chemical synthesis and potential toxicity [7]. Likewise, commonly used synthetic phenolic antioxidants, such as butylated hydroxytoluene and butylated hydroxyanisole, also entail undesirable properties, including the high volatility, instability at higher temperatures, and carcinogenic potential [8].

An alternate and safer choice may rely on natural compounds. Pigments from natural origin, such as those from bacteria are promising alternatives once bacteria can produce large amounts of pigments in short production times, and these pigments present better biodegradability, and are environmentally friendly. The production of bacterial pigments through fermentation has increased, the previous year’s high yields in pigments production with a low amount of residues can be achieved, being season independent, nontoxic, and with easy separation from cell biomass compared with pigments extracted from plants. Microorganisms can be tailored to provide high concentrations in short times by submerged or solid-state fermentation, where process parameters such as pH, temperature, carbon source, and aeration must be optimized [9,10,11,12,13,14,15]. In addition, handling of natural dyes is safer and non-carcinogenic (application and use) and these pigments also exhibit several interesting bioactive properties. For instance, UV protection, antimalarial, anticancer, antifungal, and immunosuppression are some of the properties that have been extensively reported for the red bacterial pigment prodigiosin [9,10,11,12], as well as its antibacterial activity [13,14,15]. Prodigiosin is a tripyrrole pigment which can be isolated from several microorganisms, namely *Serratia marcescens, Serratia rubidaea, Pseudomonas magneslorubra, Vibrio psychroerythrous, Vibrio gazogenes, Alteromonas rubra, Rugamonas rubra, Streptoverticillium rubrireticuli,* and *Streptomyces longisporus ruber* [10,16]. Its production has also been reported in *S. plymuthica*, the bacterial resource used in this study [17,18].

Moreover, the yellow flexirubin-type pigment, which has been applied in the treatment of chronic skin disease, eczema, and gastric ulcers, also displays a wide range of pharmacological effects, which includes a high antioxidant potential [19,20]. In this study, the flexirubin-type pigment used was produced by *Chryseobacterium shigense* isolated from lactic acid beverage [21]. Flexirubin was first isolated from *Flexibacter elegans*, and became the representative of a novel class of pigments produced by several genera of Bacteroidetes, such as *Flexibacter* sp, *Cytophaga* sp, *Sporocytophaga* sp, and *Chryseobacterium* sp, with each genera producing modified species of flexirubin [22,23,24]. An additional advantage of using pigments to add bioactivity to BC film-based material is also the color provided, which generates value-added products, eliminating unnecessary finishing steps. The coloration of BC has been a subject of several studies, with direct, acid and reactive synthetic dyes, to produce colored BC fabrics [25] with natural dyes such as anthocyanins from plants, for colorimetric pH indicator development [26], using immobilized enzymes, laccases for flavonoids polymerization in BC membranes [27], and even to produce functional foods using pigments of fungal origin [28,29]. However, to the best of our knowledge, no study has been conducted regarding functionalization of BC with pigments of bacterial origin, to obtain colored and bioactive materials.

Thus, this study aimed to functionalize BC film-based materials with the bacterial pigment prodigiosin and flexirubin-type pigment. The pigments were used to obtain multifunctional BC, with color and bioactive properties. Therefore, BC was functionalized at 40 °C, with 15% over the weight of the fiber (owf). The hydrophilicity, porosity, and mechanical properties of BC samples were evaluated as well as their bioactivity, namely the ability to inhibit the growth of pathogenic bacteria and free radical-scavenging activity, to anticipate its potential application.

## 2. Materials and Methods

### 2.1. Materials

Sodium hydroxide, peptone, nutrient agar, nutrient broth, brain heart infusion agar, hydrochloric acid, glycerol, and glucose monohydrate were purchased from Sigma-Aldrich (Sigma-Aldrich, St. Louis, MO, USA). Sodium chloride, ethanol and dipotassium hydrogen phosphate were from Fisher Chemical (Fisher Scientific, Leicestershire, UK) and Labkem (Labkem, Barcelona, Spain) provided the agar-agar. Potassium persulfate was purchased from Acros Organics (Acros Organics, Geel, Belgium), phosphate-buffered saline (PBS) was acquired from Alfa Aesar (Thermo Fisher GmbH, Kandel, Germany), and ABTS was provided by PanReac AppliChem (PanReac AppliChem, Darmstadt, Germany).

### 2.2. Bacterial Resources: Production and Recovery

#### 2.2.1. Bacterial Cellulose

The experimental procedure for the production of BC was adapted from a previous report from our work group [30]. Briefly, a BC starter culture was incubated in 10% (*v/v*) of a previously fermented medium, containing a microbial consortium of live bacteria and yeast species from a commercial Kombucha beverage (Freshness Diagonal, Lda, Montijo, Portugal). Bacteria and yeasts in the symbiotic association have complementary roles: yeasts hydrolyze the carbon sources available in the fermentation media, which are further consumed by the bacteria able to synthesize the BC [31,32,33,34,35]. Growth was carried out at 30 °C in a bioreactor with 8.25 g/L of commercial green tea, 8.25 g/L of commercial black tea, and 70 g/L glucose. After 7 days of fermentation the production was complete and the film-based materials were recovered and washed. Untreated BC was only washed with distilled water and alkaline-treated BC was submitted to a washing procedure, which was accomplished with a sodium hydroxide solution of 0.1 M, at 80 °C, during 30 min, as described for the cotton fibers procedure [36,37]. The film-based materials were then rinsed with distilled water until neutral pH conditions, and dried until constant weight was achieved.

#### 2.2.2. Prodigiosin Pigment

The prodigiosin pigment was produced by *Serratia plymuthica*, which was purchased from Leibniz Institute DSMZ-German Collection of Microorganisms and Cell Cultures (DSMZ, Braunschweig, Germany). The bacterial growth and pigment production was accomplished in peptone glycerol phosphate (PGP) medium, containing 5 g/L peptone, 2 g/L K_2_HPO_4_, 10 mL/L glycerol, and 15 g/L agar, for solid growth [18]. The fermentation was carried out at 20 °C, under 150 rpm shaking, with the absence of light. The cells were harvested at the late log phase and the pigment extraction procedure was performed accordingly with a previously reported method [38] with acidified ethanol.

#### 2.2.3. Flexirubin-Type Pigment

Flexirubin-type pigment production was accomplished with bacteria *Chryseobacterium shigense*, purchased from Leibniz Institute DSMZ-German Collection of Microorganisms and Cell Cultures (DSMZ). The bacterial growth and pigment production was achieved in nutrient medium (peptone 5 g/L; meat extract 3 g/L; agar, when necessary, 15 g/L), at 30 °C. Aeration and agitation rate were kept at 1/5 O_2_ and 200 rpm, respectively. The yellowish-orange pigment was recovered from the fermentation broth, using acetone, as previously described [22].

### 2.3. BC Functionalization with Prodigiosin and Flexirubin-Type Pigment

Datacolor AHIBA IR equipment (Datacolor company, Lawrenceville, NJ, USA) was used for the BC film-based materials functionalization with prodigiosin and flexirubin-type pigment. Alkaline-treated BC was functionalized with each pigment at 15% owf, corresponding to 5 mg/mL, during 60 min, at 40 °C. Moreover, the raise velocity was 2 °C/min, liquor ratio 1:30, and 20 rpm.

### 2.4. BC samples Characterization

#### 2.4.1. Attenuated Total Reflectance Fourier Transform Infrared Spectroscopy (ATR-FTIR)

The presence of prodigision and flexirubin-type pigment in BC samples, after the functionalization procedure, was evaluated by FTIR (Thermo-Nicolet is10 FT-IR Spectrophotometer, Waltham, MA, USA) with an attenuated total reflectance (ATR) accessory, operating in the transmission mode. The scanning range was 4000–600 cm^−1^, with a resolution of 4 cm^−1^.

#### 2.4.2. Color Characteristics

The color strength (K/S value) of dyed samples was obtained using a Datacolor 110 spectrophotometer (Datacolor company, Lawrenceville, NJ, USA), directly on the air-dried surface, and calculated by:(1)$$K/S={\left(1-R\right)}^{2}/2R$$
where R is the observed reflectance of dyed sample under the wavelength of 535 nm or 450 nm, for samples functionalized with prodigiosin or flexirubin-type pigment, respectively, K is the absorption coefficient, and S is the scattering coefficient.

Moreover, the colorimetric properties of the functionalized samples were also evaluated in terms of CIELab and CIELch values (L*, a*, b*, C*, and h), where L* indicates lightness from black to white (0 to 100, respectively), positive and negative values in a* represent redness and greenness, respectively, whilst positive and negative values in We are fine with the extension of two weeks. represent yellowness and blueness, respectively. Chroma values indicate the saturation or purity of the color and hue angle values represent 0° for redness, 90° for yellowness, 180° for greenness, and 270° for blueness, using polar coordinates [39,40]. The colorimetric properties of BC samples submitted to the alkaline washing procedure and recovered BC only washed with distilled water were also evaluated, and their whiteness index (WI) was obtained from Equation (2) [41]:(2)$$\mathrm{WI}=100-{\left[{\left(100-L^{*}\right)}^{2}+(a^{*}{}^{2}+b^{*}{}^{2})\right]}^{\frac{1}{2}}$$

The International Commission on Illumination (CIE) Chromaticity Diagram was obtained using the GOCIE software [42].

#### 2.4.3. Water Contact Angle Measurements

The contact angle measurements of BC samples were carried out with a Dataphysics Contact Angle System OCAH-200 apparatus (DataPhysics Instruments, Filderstadt, Germany), operating at 25 °C. Deionized water was used as reference fluid, droplets of 4 µL were placed at different locations on the material surface in each sample. The measurement was reproduced 10 times for each sample and the average value reported.

#### 2.4.4. Porosity

The porosity of the samples was assessed by liquid displacement, using ethanol. The samples were weighted ($W_{s}$) as well as a measuring cylinder, for each sample, filled with ethanol ($W_{1}$). In order to aid the ethanol penetration into the pores, the samples placed in the cylinders were sonicated in a water bath at 30 °C, for 40 min. Afterwards, the cylinders, containing the samples, were refilled with ethanol and weighed ($W_{2}$), then the samples were removed, and the cylinders weighed again ($W_{3}$). The porosity (ε) of the samples was calculated using the following equation, as described elsewhere [43]:(3)$$\varepsilon \;\left(\%\right)=\left(W_{2}-W_{3}-W_{s}\right)/\left(W_{1}-W_{3}\right)\times 100$$

#### 2.4.5. Mechanical Testing

The mechanical performance of BC samples, in dry form, was evaluated by tensile tests (tensile strength, Young’s modulus and elongation at break were evaluated) performed on a Universal tensile test machine (Adamel Lhomargy Division d’Instruments Model DY-35, Roissy en Brie, France). The apparatus was equipped with a 100 N static load cell. The samples were cut into strip-shaped specimens of 5 mm width and 30 mm long, with an average thickness of 0.04 mm and 0.4 mm (Adamel Lhomargy, m120, Roissy en Brie, France), for BC samples alkaline-treated or untreated BC, respectively. Samples were stored in desiccators for at least 24 h before testing and the tensile testing was undertaken at room temperature, according to standard ASTM D3039/D3039M (standard test method for tensile properties of polymer matrix composite materials). The testing was performed for at least five specimens, at 1 mm/min, as previously reported [44].

### 2.5. Antibacterial Activity

The ability BC and functionalized BC samples to inhibit *S. aureus* (ATTC 6538) and *P. aeruginosa* (PA25) growth was tested through ASTM E2180-07 (standard test method for determining the activity of incorporated antimicrobial agent(s) in polymeric or hydrophobic materials), for a more even contact of the inoculum with the BC samples. Sterile agar slurries were prepared with 0.85 (*w*/*v*) NaCl and 0.3 (*w*/*v*) agar-agar and 1 mL of the bacterial suspensions (10^8^ CFU/mL), which were prepared with an overnight stationary liquid culture, were added to 100 mL of sterile agar slurries. The bacterial suspensions placed in the agar slurries were then inoculated over the functionalized BC samples and the alkaline-treated BC, which acted as a control. The antibacterial activity was assessed immediately after inoculum application (T0h), and after 24 h (T24h) in contact with the agar slurries, at 37 °C. Serial dilutions of the agar slurries recovered at T0h and T24h were carried out with 0.85 (*w*/*v*) NaCl and spread on Nutrient Agar plates, which were incubated at 37 °C for 24 h, in order to determine CFU/mL. Then, the percentage of bacterial reduction (%R) was calculated accordingly with Equation (4):(4)Percentage Reduction (%R)=((C−S)/C)×100
where S represents the number of CFUs obtained with the BC-functionalized samples and C is the CFUs of bacteria recovered from alkaline-treated BC control.

### 2.6. Antioxidant Activity

ABTS radical decolorization assay was used to determine the functionalized BC samples scavenging capacity of ABTS radical cation (ABTS^+^), as previously described [45,46]. Briefly, ABTS stock solution (7 mM) was mixed with 2.45 mM potassium persulfate (final concentration), and the mixture was kept in the dark at room temperature for 12–16 h, to produce ABTS^+^. Then, the produced ABTS^+^ solution was diluted with phosphate buffer (0.1 M, pH 7.4) to reach an absorbance of 0.700 ± 0.025, at 734 nm. The reaction of each sample (10 mg) with ABTS^+^ solution (10 mL) occurred in the dark, during 30 min, and the scavenging capability of ABTS^+^ at 734 nm was calculated using Equation (5):(5)$$\mathrm{Antioxidant}\;\mathrm{Activity}\;\left(\%\right)=\frac{A_{\mathrm{Control}}-A_{\mathrm{sample}}}{A_{\mathrm{control}}}\times 100$$
where A_control_ is ABTS^+^ initial absorbance (contains all the test reagents without the test samples) and A_sample_ is the absorbance of the remaining ABTS^+^ in the presence of the tested sample.

### 2.7. Statistical Analysis

Each experiment was performed at least three times, unless otherwise stated. The statistical analysis performed, using GraphPad Prism 6 software, was a one-way analysis of variance (ANOVA), followed by Tukey’s multiple comparisons test. A *p*-value below 0.05 was considered statistically significant. Data are expressed as mean ± standard deviation (SD).

## 3. Results and Discussion

### 3.1. BC films Functionalization with Prodigiosin and Flexirubin-Type Pigment

BC functionalization with prodigiosin and flexirubin-type pigment was intended to yield BC films with multifunctional properties. On one hand, the purpose was to obtain colored BC, since color plays a crucial role in our perception and appeal of certain items and, on the other hand, the aim was also to take advantage of the outmost important biological activities of these bacterial pigments, such as the antimicrobial and antioxidant properties. The process of BC functionalization was designed to obtain the desirable properties with cost-effectiveness and low environmental impact. Lower pigment concentrations were tested at 40 °C, during 60 min, 20 rpm, with a raise velocity of 2 °C/min, and liquor ratio 1:30, namely 5% owf (1.67 mg/mL) and 10% owf (3.33 mg/mL), data not shown. Moreover, the temperature was maintained at 40 °C during all the functionalization procedures since microbial pigments possess lower thermostability compared with synthetic dyestuff and higher temperatures require high energy amounts which, ultimately, translates into higher costs and environmental impact [47,48]. Nonetheless, it was only at 15% owf (5 mg/mL) that BC presented good coloration for the naked eye and higher color strength, 1.0 ± 0.0 for BC functionalized with prodigiosin and 0.4 ± 0.0 for BC functionalized with flexirubin-type pigment (Table 1). Figure 1 exhibits the BC samples CIE points calculated using the GOCIE software, from the spectra obtained from each sample, using Datacolor 110 spectrophotometer [42]. This chromaticity chart may assist in establishing a quantitative context for the visualization of the samples color distribution.

Furthermore, BC was dyed pinkish by the microbial pigment prodigiosin, as confirmed by the hue angle value of 2.6 ± 0.7°, and indicated by the a* positive value of 22.4 ± 0.1 (Table 1). Even though studies with prodigiosin incorporated in different textile substrates report a wide range of K/S values (from 0.3 to 3.2), the a* positive value and hue angle value closer to 0° are common denominators between the reports. Moreover, K/S values may vary significantly with several factors: pH, dyeing temperature, percentage of ethanol, dyeing time, and the substrate, probably due to fibers affinity [9,13,18,47].

Concerning the BC samples functionalized with flexirubin-type pigment, the yellowness obtained was verified by the hue angle value of 82.6 ± 1.5°, and a b* positive value of 24.18 ± 0.5 was also observed (Table 1). Likewise, previous reports regarding the colorimetric properties of flexirubin-type pigments extracted from different *Chryseobacterium* strains, also reveal hue angle values closer to 90°, which represents colors in the yellow region [22,49,50].

Additionally, the colorimetric properties of BC film-based materials before and after alkaline treatment were also determined to evaluate their whiteness index. As reported in Table 1, the BC whiteness index before the alkaline procedure was 40.3 ± 1.9, and BC pellicles displayed a brown coloration after recovery and washing with only distilled water. The brownish color observed can be attributed to fermentation medium components and cellular debris, such as proteins and nucleic acids, from bacteria that remained in the BC materials. Indeed, previous reports showed numerous oval bacteria on the surface of recovered Bacterial Cellulose, as well as the dark BC coloration due to the resulting cellular debris [41,44]. After the alkaline treatment, the BC materials exhibited a considerably higher whiteness index of 79.0 ± 1.2, since the washing procedure enables the hydrolysis and removal of impurities [44]. Moreover, the higher whiteness index displayed by BC after the alkaline procedure contributes to the uniform coloration and the bright colors observed after the functionalization with the bacterial pigments [41].

An FTIR analysis was performed to evaluate the pigments incorporation in BC samples and the BC spectra before and after alkaline treatment were also analyzed. The major differences observed in the spectra, as seen in Figure 2, were obtained for untreated and alkaline-treated BC, where the hydrogen-bonded OH stretching at 3278 cm^−1^ is shifted to a higher wavenumber, 3339 cm^−1^, and a shift in the absorption band area was also observed, which can be related with the disappearance of the N-H stretch (from proteins and amino acids removed during the alkaline treatment). The peak at 1731 cm^−1^, assigned for C=O group in proteins and lipids, became almost imperceptible after the purification step [44,51,52]. As can be seen in Figure 2, the spectra obtained for alkaline-treated BC and BC functionalized with prodigiosin are very similar, which can be attributed to the low amounts of prodigiosin incorporated. Even so, increased absorption bands intensity is especially noticeable at 3341 cm^−1^, which can be attributed to the N-H stretch of prodigiosin; the higher difference was identified at 2980 and 2889 cm^−1^, due to asymmetrical and symmetrical stretching of methylene groups [53,54,55]. The spectra obtained for BC functionalized with flexirubin-type pigment also showed different peak intensities at specific wavenumbers, namely the increased intensity at 1732 cm^−^^1^, assigned to asymmetrical stretching of C=O bond of the ester linking the phenol with the polyunsaturated chain of the pigment, and the peak at 1464 cm^−1^ which is characteristic of the aromatic ring of the phenol [50,56]. Overall, the differences in spectra observed indicate the incorporation of both pigments in the BC materials. 

### 3.2. BC Film-Based Materials Characterization

Once the BC samples were subjected to NaOH alkaline treatment, prior to functionalization with the bacterial pigments, the WCA measurements of the functionalized materials were compared with untreated BC and alkaline-treated BC.

The NaOH alkaline treatment was used to eliminate cellular debris and medium components [37] but it also affected BC intrinsic properties. Thus, as can be seen in Figure 3A, the untreated BC WCA was 20.80 ± 1.65°, which indicates an almost very hydrophilic material [57], as expected of BC. However, after the alkaline treatment, the WCA of BC slightly increases to 47.07 ± 5.51°, characteristic of moderate hydrophilic substrates (40° < WCA < 70°) [57]. During the alkaline treatment, the surface area of BC increases, due to swelling of the fibers, the crystallinity index decreases, an intermolecular rearrangement of individual chains occurs, and the surface of the films usually follows a decrease in hydrophilicity [58], which explains the increased WCA of alkaline-treated BC. After pigments incorporation, the BC films maintained a moderate hydrophilic character, with 47.43 ± 7.13° for BC functionalized with flexirubin-type pigment, and 49.20 ± 8.84° WCA for BC functionalized with prodigiosin. The difference between alkaline-treated BC before and after the functionalization procedure was not significant.

Moreover, the hydrophilicity decrease of BC is a desired effect for the application of BC materials in a wider range of applications, and further improvements can also be performed, for example with an additional hydrophobic finishing step, as already reported by our work group, where hydrophobic BC was obtained with minimal quantities of hydrophobic finishing agents [30]. Other strategies to yield hydrophobic BC can also be used. Leal et al. recently reported the use of oxygen plasma treatment and chemical vapor deposition [52]. Another option can be acetylation, which is a very common reaction that introduces an acetyl functional group onto the BC surface, the acetyl groups react with the BC surface hydroxyl groups, changing the hydrophilicity of the BC surface. The degree of substitution, and consequently, the BC hydrophilicity, can be controlled by performing heterogeneous acetylation to improve its properties and preserve its morphology with hydrophobic surface characteristics [59,60].

BC porosity is influenced by several factors during its production and recovery, namely, cultivation time, inoculation volume, post-treatment, and drying methods [61,62]; nonetheless, the values reported for BC are porosity values superior to 85% [63]. In fact, as can be seen in Figure 3B, the BC porosity obtained in our work (89.13 ± 1.78%) is in accordance with previous reports, including the report by Yin and collaborators, where BC was also subjected to an alkaline post-treatment [64]. After the functionalization with prodigiosin and flexirubin-type pigment, the BC porosity results obtained (85.19 ± 1.90% and 84.51 ± 5.06%) suggest that the pigments incorporation does not affect BC porosity, since the differences obtained were not significant.

Additionally, the production process and fermentation conditions also affect the BC high degree of polymerization, crystallinity index, and its web-like network structure, which directly influence BC mechanical properties [1,2,3,65].

In this study, the mechanical performance was evaluated for the untreated BC, alkaline-treated BC, and BC functionalized with prodigiosin and flexirubin-type pigment, the results are summarized in Figure 4. Alkaline-treated BC presented higher Young’s modulus and tensile strength, but lower elongation at break, compared with untreated BC, indicating that alkaline-treated BC was more brittle, with an elongation at break of 8.85 ± 1.20%, compared with untreated BC, which exhibited higher deformation, with a superior elongation at break of 162.95 ± 10.25%, and the ability to absorb more energy under tensile loading (Figure 4A,B). This tendency was expected due to the removal of unwanted debris from the membranes by alkali treatment, which allows the BC fibrils to better interact with each other, increasing the intrinsic hydrogen bonding found within the BC sheet and exhibiting lower ductility [44,66]. Subsequently, the fibrils fail to realign easily when subjected to a tensile load, which leads to BC fracture with little plastic deformation, as can be observed in Figure 4B.

Furthermore, the process of BC functionalization with prodigision and flexirubin-type pigment does not affect significantly the BC mechanical properties, as can be seen in Figure 4, with samples displaying similar ductility such as alkaline-treated BC prior the functionalization procedure.

Nonetheless, for specific applications of the produced colored materials, where more elasticity or an overall higher mechanical performance would be desirable, introduction of a moisturizer [67] or chemical modifications, such as oxidation [68,69,70,71], acetylation [72,73], silylation [74] or etherification [75], can also be performed.

### 3.3. Bioactivities of the Functionalized BC Materials: Antibacterial and Antioxidant Activity Evaluation

The Minimum Inhibitory Concentration (MIC) reported in some studies with prodigiosin against the *S. aureus* is less than 10 µg/mL, but higher concentrations are required for considerable growth inhibition of *P. aeruginosa* [14,18]. Nonetheless, due to the relatively high concentration of prodigiosin used in this study to produce the colored BC films, 5 mg/mL, a high inhibitory effect against the tested pathogenic bacteria was expected. In fact, the BC samples functionalized with prodigiosin, after 24 h of contact, showed a total reduction in *S. aureus* and *P. aeruginosa* viability of 97.89 ± 0.60% and 85.12 ± 0.17%, respectively (Figure 5A).

The higher inhibitory effect of prodigiosin observed on Gram positive bacteria, *S. aureus*, than in Gram negative bacteria, *P. aeruginosa*, is in accordance with previous reports [10]. It has also been previously proposed that prodigiosin acts as a chaotropic stressor, targeting and disrupting the bacterial plasma membrane, as its primary antimicrobial activity mode-of-action [76,77]. Despite the fact that, in this study, the antibacterial activity of prodigiosin was only tested against two pathogenic bacteria, there are countless studies reporting a broader spectrum of its antibacterial activity [77], which enhances the suitability of the produced colored material for countless applications, for example in the active food packaging field, where besides color, the antimicrobial activity also plays a crucial role to enhance the shelf life of food products [78], or for medical applications, and even for the development of innovative and functional textile materials, without requiring additional finishing steps [13,79].

On the other hand, BC samples functionalized with flexirubin-type pigment exhibited weaker and much inferior antibacterial activity against *S. aureus*, with only a total reduction of 66.82 ± 4.42%, and besides, no inhibition of *P. aeruginosa* growth was observed (Figure 5A).

Nonetheless, flexirubin-type pigments exhibit other relevant bioactive properties, such as the antioxidant ability, which was also assessed. ABTS^+^ radical scavenging assay is an electron transfer-based assay widely used for the assessment of the antioxidant capacities of natural products [80,81,82,83]. This spectrophotometric technique is based on ability of compounds to donate hydrogen to free radicals and thus the antioxidants reduce the ABTS^+^ radical to a colorless compound [84]. In this study, the scavenging ability of pigment-functionalized BC samples on the ABTS^+^ radical, was evaluated to determine the effect of adding prodigiosin and flexirubin-type pigment on the antioxidant activity of the BC samples. Nonetheless, BC samples without functionalization with either pigment exhibited some antioxidant activity, 21.86 ± 1.53% (Figure 4B), which can be attributed to the fermentation conditions used in this study to produce the BC materials. The polyphenols released from black and green tea mixture used in the fermentation medium are known for their ability to scavenge free radicals [85]. As a matter of fact, [86] and collaborators recently reported an 8 to 10 times higher antioxidant activity for BC produced with green tea compared with BC produced in a defined growth medium such as Hestrin–Schramm [86]. Moreover, the results showed that highest ABTS^+^ scavenging activity was obtained for BC samples functionalized with flexirubin-type pigment, at 38.96 ± 0.49% (Figure 5B). The antioxidant activity of the flexirubin-type pigment, which can be attributed to the hydrogen atom transfer from the phenol, has been previously determined through different assays, besides the radical scavenging activities, namely lipid peroxide inhibition and ferrous chelating ability [20], which support the significance of flexirubin-type pigment as a powerful antioxidant, relevant for a wide range of applications.

The radical scavenging capacity against ABTS^+^ was also measured to determine the anti-oxidative capacity of prodigiosin-functionalized BC samples; however, the antioxidant activity obtained was not significantly different from the antioxidant activity of uncolored BC samples, which can be a result of the concentration dependency of prodigiosin antioxidant activity, as previously reported [87,88].

## 4. Conclusions

In this study, a sustainable synthesis and functionalization of BC film-based materials using renewable sources and the bacterial pigment prodigiosin and flexirubin-type pigment produced colored and bioactive BC materials, without adding any chemical mordant or finishing agents. The results showed that the functionalization process at 15% owf, 40 °C during 60 min, was sufficient to obtain BC colored materials which were also able to inhibit the growth of pathogenic bacteria such as *S. aureus* and *P. aeruginosa*, with the prodigiosin pigment, and with scavenging capacity against ABTS^+^ radical, with the flexirubin-type pigment. Moreover, BC hydrophilicity, porosity, and mechanical performance, were preserved after both pigments’ incorporation, and present good perspectives for medical or other applications where antibacterial and antioxidant functions are required. Thus, the multifunctional colored and bioactive materials produced may have a wide range of applications, including in the food packaging industry, textile, and even medical applications.

A possible medical application of the BC materials herein developed to take full advantage of films’ properties, namely the colors provided by pigments’ incorporation, and the antibacterial activity to avoid wound infection, can be the development of intelligent wound dressing materials, benefiting from the color change in the bacterial pigments to alkaline pH normally associated with wound infection.

## Figures and Tables

**Figure 1 materials-15-02069-f001:**
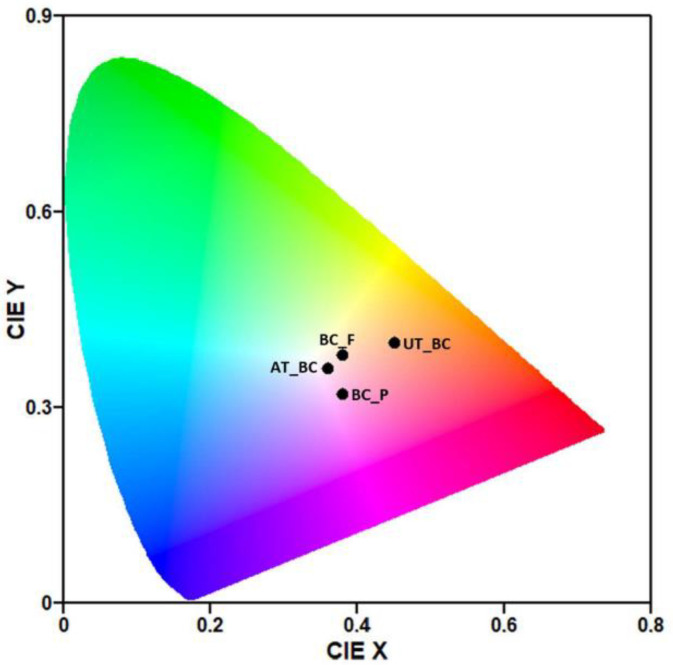
CIE Chromaticity diagram obtained using GOCIE software [42], from the sample’s spectra provided by Datacolor 110 spectrophotometer. The diagram allows the perception of the samples color from the distributions of wavelengths in the electromagnetic visible spectrum.

**Figure 2 materials-15-02069-f002:**
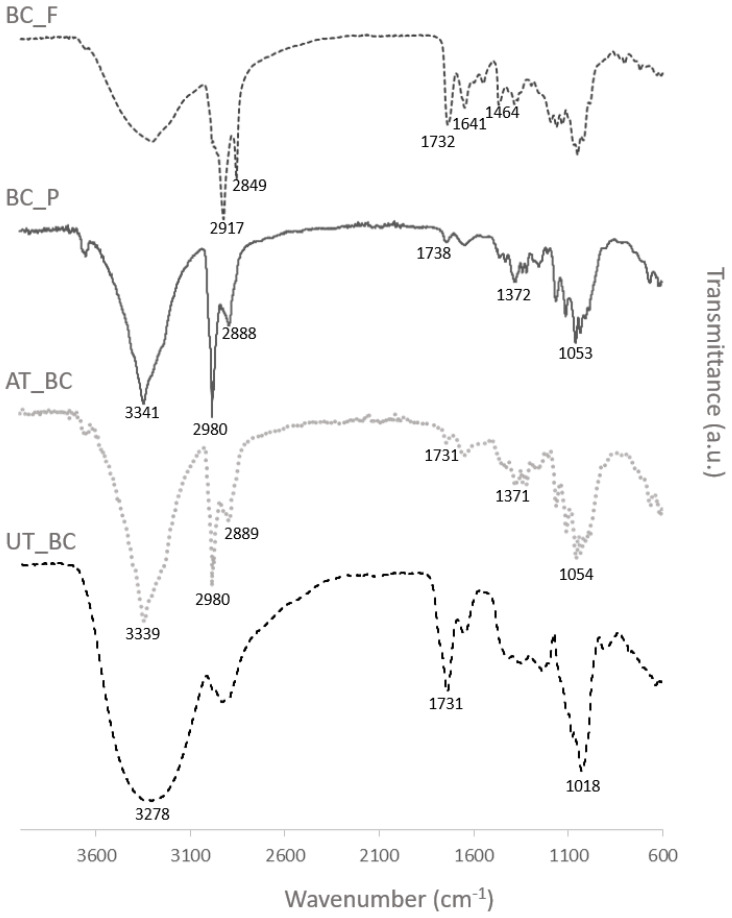
FTIR spectra of BC samples: untreated BC (UT_BC), alkaline-Treated BC (AT_BC), BC after functionalization with prodigiosin (BC_P), and BC after functionalization with flexirubin-type pigment (BC_F).

**Figure 3 materials-15-02069-f003:**
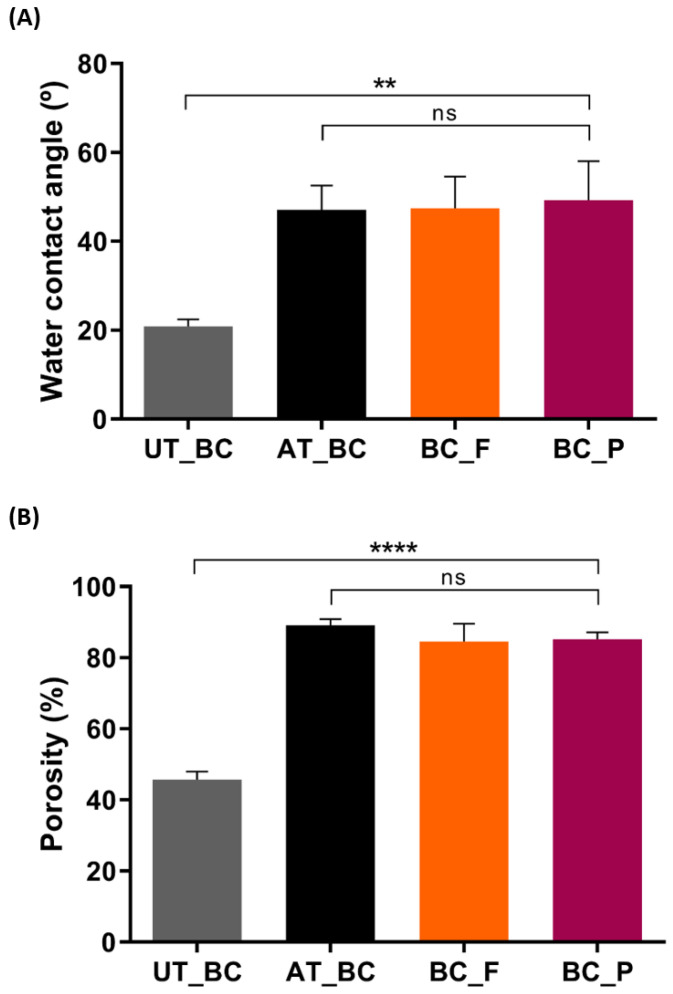
(**A**) Water contact angle and (**B**) porosity of BC samples: untreated BC (UT_BC), alkaline-treated BC (AT_BC), BC after functionalization with prodigiosin (BC_P), and BC after functionalization with flexirubin-type pigment (BC_F). (Data are presented as mean ± SD, ns indicates non-significant, ** *p* < 0.01, and **** *p* < 0.0001).

**Figure 4 materials-15-02069-f004:**
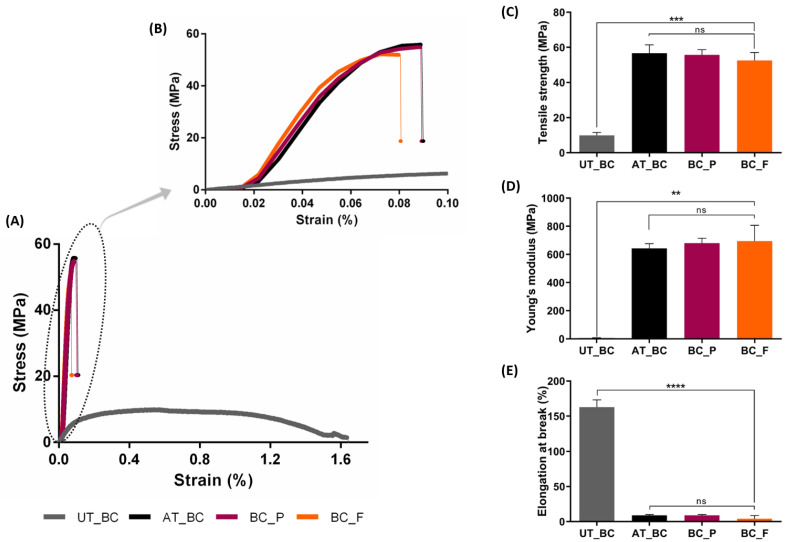
Summary of mechanical performance of untreated BC (UT_BC), alkaline-treated BC (AT_BC), BC functionalized with prodigiosin (BC_P), and BC functionalized with flexirubin-type pigment (BC_F): (**A**) tensile stress-strain curves; (**B**) close-up view of tensile stress-strain curves of AT_BC, BC_P, and BC_F; (**C**) tensile strength, (**D**) Young’s modulus, and (**E**) elongation at break of BC samples. (Data are presented as mean ± SD, ns indicates non-significant, ** *p* < 0.01, *** *p* < 0.001, and **** *p* < 0.0001).

**Figure 5 materials-15-02069-f005:**
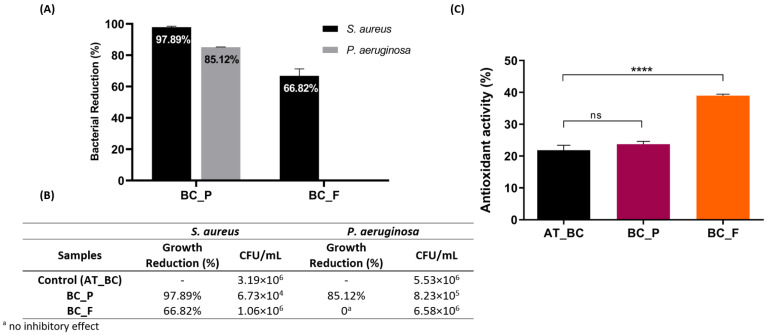
Antibacterial and antioxidant activity evaluation: (**A**) growth inhibition by BC functionalized samples, with prodigiosin (BC_P) and flexirubin-type pigment (BC_F), against pathogenic bacteria *S. aureus* and *P. aeruginosa*; (**B**) correspondence between the percentage of bacterial reduction and the respective CFU/mL; (**C**) percentage of radical ABTS scavenging activity of BC samples (alkaline-treated BC, AT_BC, BC after functionalization with prodigiosin, BC_P, and BC after functionalization with flexirubin-type pigment BC_F). (Data are presented as mean ± SD, ns indicates non-significant, and **** *p* < 0.0001).

**Table 1 materials-15-02069-t001:** Colorimetric properties of BC samples, with whiteness index of BC before functionalization (untreated BC, UT_BC, and alkaline-treated BC, AT_BC) and color strength (K/S value) of BC samples after functionalization with prodigiosin (BC_P) and flexirubin-type pigment (BC_F).

		UT_BC	AT_BC	BC_P	BC_F
Apparent color					
Reflectance (%R)		-	-	26.8 ± 0.1	42.7 ± 1.0
Color strength (K/S)		-	-	1.0 ± 0.0	0.4 ± 0.0
Whiteness Index (WI)		40.3 ± 1.9	79.0 ± 1.2	-	-
CIELab and CIELch colorimetric parameters	L*	51.5 ± 3.5	83.6 ± 1.0	71.1 ± 0.4	86.1 ± 0.4
	a*	14.8 ± 0.6	1.5 ± 0.4	22.4 ± 0.1	3.1 ± 0.7
	b*	31.5 ± 2.1	13.1 ± 0.7	1.0 ± 0.3	24.1 ± 0.5
	Chroma (C*)	34.8 ± 1.7	13.2 ± 0.7	22.4 ± 0.1	24.3 ± 0.6
	Hue angle (°)	64.7 ± 2.4	83.4 ± 1.2	2.6 ± 0.7	82.6 ± 1.5

## Data Availability

Data that support the findings of this study are included in the article.

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
