# Peer review of "Characterization of Bioactive Colored Materials Produced from Bacterial Cellulose and Bacterial Pigments"

_materials, 2022, doi:10.3390/ma15062069_

Round 1

Reviewer 1 Report

  1. The introduction to work is sufficient and provides a good background for the set goal.
  2. The literature review is sufficient. The literature list is quite extensive.
  3. The presented research methodology is correct. The authors demonstrated good researcher's skills.
  4. It would be worth supplementing the work with information which of the research was carried out according to some standards.
  5. The presentation of the results is correct. The figures are a bit too small (size). The interpretation is correct and supported by right conclusions.
  6. The layout of the tables is clear and legible, they are well described in the main text.
  7. The final conclusions are correctly formulated. The authors emphasize the practical use of the material. It has this significant value added to all the work.

Author Response

Response: The authors greatly acknowledge the reviewer’s comments and suggestions. Following the reviewer's recommendations, the work was supplemented with information regarding standards (Page 5, lines 205,206,211-213). The figures size in the revised manuscript was also increased.

Reviewer 2 Report

The manuscript entitled Sustainable and bioactive colored materials produced from bacterial cellulose and bacterial pigments presents information related to the synthesis of microbial biofilms incorporated with bacterial pigments, its characterization, and evaluation of bioactivities. The manuscript presents several issues that authors must attend to prior to acceptance. Below are the comments.

-The title does not represent the content of the manuscript. The title must be Characterization of microbial (or the name of starter culture) cellulose biofilm incorporated with bacterial pigments.

-Abstract section. it is important to mention how was developed bacterial cellulose films and which microorganisms produced bacterial pigments.

-Introduction. Provide information related to the importance of using Prodigiosin. Also, depict what was the rationale of incorporating Prodigiosin and Flexirubin into the biofilm.

-Materials and methods. Section 2.2.1. It is important that the authors provide detailed information related to the BC starter culture. What was the microorganism used for the synthesis of biofilm? The authors cited their previous work, however, the document did not mention any information about the microbe or microbes.

-Results and discussion section. Lines 227-239 must be included in the introduction section.

- Section 3.3 and figure 5A. How was estimated the antibacterial activity percentage since the authors determined the CFU/mL? Place in the materials and methods section the equation used. What was the control treatment?

Author Response

Reviewer #2: The manuscript entitled Sustainable and bioactive colored materials produced from bacterial cellulose and bacterial pigments presents information related to the synthesis of microbial biofilms incorporated with bacterial pigments, its characterization, and evaluation of bioactivities. The manuscript presents several issues that authors must attend to prior to acceptance. Below are the comments.

1-The title does not represent the content of the manuscript. The title must be Characterization of microbial (or the name of starter culture) cellulose biofilm incorporated with bacterial pigments.

Response 1: The authors greatly acknowledge the suggestion that we have partially agreed. Accordingly, the title was modified to: Characterization of bioactive colored materials produced from bacterial cellulose and bacterial pigments, because not only characterization as development/production is provided.

2-Abstract section. it is important to mention how was developed bacterial cellulose films and which microorganisms produced bacterial pigments.

Response 2: The authors agree and greatly acknowledge the suggestion. Accordingly, the information regarding the bacterial cellulose films development from a microbial consortium of bacteria and yeasts, as well as the name of the bacterial pigment producers, were added to the abstract (Page 1, line 13-15).

3-Introduction. Provide information related to the importance of using Prodigiosin. Also, depict what was the rationale of incorporating Prodigiosin and Flexirubin into the biofilm.

Response 3: The importance of using prodigiosin is majorly ascribed to its antibacterial properties and safer use, when compared with other antimicrobial agents used to endow Bacterial Cellulose with antimicrobial properties, as referred to in the Introduction (page 2, line 45-47). The rationale of incorporating Prodigiosin and Flexirubin pigments in Bacterial cellulose was to produce multifunctional film-based materials with color and bioactive properties, which could find application in several fields (Page 2, line 91).

4-Materials and methods. Section 2.2.1. It is important that the authors provide detailed information related to the BC starter culture. What was the microorganism used for the synthesis of biofilm? The authors cited their previous work, however, the document did not mention any information about the microbe or microbes.

Response 4: The authors greatly acknowledge your question, and we apologize for the incomplete information. The Bacterial Cellulose fermentation was carried out using 10% (V/V) of a previous fermented culture containing a microbial consortium of live bacteria and yeast species from a commercial Kombucha beverage obtained from a wholesale distributor (Freshness Diagonal, Lda, Montijo, Portugal). This information was added to the revised manuscript (Page 3, line 108-110).

5-Results and discussion section. Lines 227-239 must be included in the introduction section.

Response 5: The authors acknowledge the suggestion and the text was transferred from the Results and discussion section to the Introduction of the revised manuscript (Page 2, line 66-70, 73-78).

6- Section 3.3 and figure 5A. How was estimated the antibacterial activity percentage since the authors determined the CFU/mL? Place in the materials and methods section the equation used. What was the control treatment?

Response 6: The authors agree with the reviewer, the equation used was missing from the Materials and methods section, it was added to the revised manuscript (Page 5, line 222-229). The Control used was alkaline-treated Bacterial Cellulose, without bacterial pigments functionalization, since Bacterial Cellulose has no inherent antibacterial activity. This information was also added to the revised manuscript (Page 5, line 217,218).

Reviewer 3 Report

Overall, the manuscript materials-1631483, involving the production of bacterial cellulose, its dyeing with microbe-derived pigments and its optical and mechanical characterization, is well structured, clear, endowed with enough experimental detail without being prolix, rich in discussion, and pertinent. Here are some suggestions to improve its quality.

  • Abbreviations: It is good practice to avoid abbreviations in the abstract unless really necessary, and also to define them in their first appearance in the text, beginning with the introduction, so the abstract could, let us say, stand alone. Hence, in the case of "owf", the authors should use its full name in the abstract and define it in the introduction — where, in the current form of the manuscript, only the abbreviation is given.

  • Line 90 -> Please give the full name for Panreac (PanReac AppliChem).

  • Lines 128-129 -> The word "Analysis" is not necessary. Also, the BC to be functionalized... Was it the untreated one or the alkaline one?

  • Lines 174-177 -> Providing 4 equations for one magnitude, and considering that a reference is given to support this calculation method, is not necessary. The last one, i.e., the second part or corollary of equation 5, should suffice. Readers will check ref. 33 if in doubt.

  • The two first paragraphs of "Results and Discussion" are not related to the results of this work. They should be drastically shortened.

  • Figure 1 uses the comma as decimal separator.

  • Confidence intervals show too many figures. Not all of them are significant, for sure. For instance, "89.13 ± 1.78" does not convey more significant information than "89.1 ± 1.8".

  • Paragraph starting @line 304 -> The authors should keep the word "peak" when absorption for a certain vibration takes place at a narrow wavelength/wavenumber. The one at 2980 cm-1, for instance. Yet, when they are broad, we should talk about absorption bands.

Also on FTIR, "1732 and 1641 cm-1, assigned to asymmetrical and symmetrical stretching of C=O bond of the ester linking the phenol..."

1732 corresponds to the asymmetrical stretching of ester's C=O, I agree. Yet, 1641 seems unlikely for the corresponding symmetrical vibration. Could not it be a carboxylate group, maybe resulting from some hydrolysis?

  • Please correct me if I am wrong, but there are no reasons for black & white printing in "Materials". Figures 3-5 would be more appealing and even easier to understand in color than in grayscale.

  • Although the manuscript is generally well-written and perfectly understandable by anyone, I have some suggestions to improve writing, particularly to avoid reiteration.

"bacterial pigments prodigiosin, and flexirubin-type pigment" > "bacterial pigment prodigiosin and a flexirubin-type pigment". While flexirubin and flexirubin-like compounds are also, generally speaking, of bacterial origin, the second appearance of "pigment" is redundant with the plural form.

A similar issue at the end of the abstract: "application in several fields, especially in medical applications." "Especially for medical purposes", for instance.

Likewise, "Pigments from natural origin such as bacterial pigments" > "Pigments from natural origin, such as those from bacteria,..."

Please seek and correct more cases of repetitive speech along the article.

Author Response

Reviewer #3: Overall, the manuscript materials-1631483, involving the production of bacterial cellulose, its dyeing with microbe-derived pigments and its optical and mechanical characterization, is well structured, clear, endowed with enough experimental detail without being prolix, rich in discussion, and pertinent. Here are some suggestions to improve its quality.

1- Abbreviations: It is good practice to avoid abbreviations in the abstract unless really necessary, and also to define them in their first appearance in the text, beginning with the introduction, so the abstract could, let us say, stand alone. Hence, in the case of "owf", the authors should use its full name in the abstract and define it in the introduction — where, in the current form of the manuscript, only the abbreviation is given.

Response 1: The authors agree and greatly acknowledge the suggestion. Accordingly, the full name was used in the abstract (Page 1, line 16) and was also defined in the introduction (Page 2, line 92).

2-Line 90 -> Please give the full name for Panreac (PanReac AppliChem).

Response 2: The authors agree and thank the reviewer for the suggestion, the full name was inserted (Page 3, line 103).

3- Lines 128-129 -> The word "Analysis" is not necessary. Also, the BC to be functionalized... Was it the untreated one or the alkaline one?

Response 3: The authors accept and appreciate the suggestion made, the word “Analysis” was removed (Page 3, line 144). The BC used in the functionalization procedure with the bacterial pigments was alkaline-treated BC, and that information was also added (Page 3, line 139).

4- Lines 174-177 -> Providing 4 equations for one magnitude, and considering that a reference is given to support this calculation method, is not necessary. The last one, i.e., the second part or corollary of equation 5, should suffice. Readers will check ref. 33 if in doubt.

Response 4: The authors agree and greatly acknowledge the suggestion, the unnecessary equations were removed from the manuscript (Page 5, line 189-195).

5- The two first paragraphs of "Results and Discussion" are not related to the results of this work. They should be drastically shortened.

Response 5: The authors agree and thank the reviewer for the suggestion. Accordingly, the "Results and Discussion" section has been shortened (Page 6, line 254-273).

6-Figure 1 uses the comma as decimal separator.

Response 6: Figure 1 was adjusted in the revised manuscript (Figure 1, page 7).

7-Confidence intervals show too many figures. Not all of them are significant, for sure. For instance, "89.13 ± 1.78" does not convey more significant information than "89.1 ± 1.8".

Response 7: The authors accepted the suggestion and the values in table 1 were updated, as well as the values reported in the text, to show less figures (Page 7 and 8, line 288-327).

8-Paragraph starting @line 304 -> The authors should keep the word "peak" when absorption for a certain vibration takes place at a narrow wavelength/wavenumber. The one at 2980 cm-1, for instance. Yet, when they are broad, we should talk about absorption bands. Also on FTIR, "1732 and 1641 cm-1, assigned to asymmetrical and symmetrical stretching of C=O bond of the ester linking the phenol..."1732 corresponds to the asymmetrical stretching of ester's C=O, I agree. Yet, 1641 seems unlikely for the corresponding symmetrical vibration. Could not it be a carboxylate group, maybe resulting from some hydrolysis?

Response 8: The authors agree with the suggestion and have improved the text, addressing absorption bands/peaks accordingly with their broad or narrow wavelength/wavenumber, respectively (Page 8, line 335,341). We also agree with the reviewer suggestion regarding the 1641 cm-1 absorption peak, it could be a carboxylate group, the information was corrected as the reviewer can check in the revised manuscript (Page 9, line 346-347).

9-Please correct me if I am wrong, but there are no reasons for black & white printing in "Materials". Figures 3-5 would be more appealing and even easier to understand in color than in grayscale.

Response 9: We agree and appreciate the reviewer’ suggestion, Figures 3-5 were updated in the revised manuscript (Pages 10,12, and 13).

10-Although the manuscript is generally well-written and perfectly understandable by anyone, I have some suggestions to improve writing, particularly to avoid reiteration. "bacterial pigments prodigiosin, and flexirubin-type pigment" > "bacterial pigment prodigiosin and a flexirubin-type pigment". While flexirubin and flexirubin-like compounds are also, generally speaking, of bacterial origin, the second appearance of "pigment" is redundant with the plural form. A similar issue at the end of the abstract: "application in several fields, especially in medical applications." "Especially for medical purposes", for instance. Likewise, "Pigments from natural origin such as

bacterial pigments" > "Pigments from natural origin, such as those from bacteria,..." Please seek and correct more cases of repetitive speech along the article.

Response 12: The manuscript was thoroughly checked and the repetitive speech along the article was eliminated to improve the quality of the manuscript (Page 1, lines 14, 27; Page 2, lines 52,90; Page 14, line 494).

Round 2

Reviewer 2 Report

The manuscript has been improved. However, there still are issues that authors must clarify. Below are the comments.

-I am recommending major revisions because it is not clear the source of the biofilm. So, since the title, the information is not clear. In the title, the authors stated biofilm is produced by bacteria, but in the reviewer responses and in the corrected manuscript they mentioned it is obtained from a consortium of bacteria and yeast. They mentioned they used commercial Kombucha starter culture. Are the microbes in the starter culture identified? How authors can assure the biofilm is produced only by bacteria?

-Figure 5 is not clear. There are overlapped images. 

-Figure 5A must be displayed by showing bars with the corresponding standard deviation for each treatment. Also, show the scale on Y-axis. Authors must include the results for the control treatment and carry out a means comparison test.

- It could be interesting that authors provide good-quality microscopic images (if possible SEM) showing the differences in biofilm structures among the treatments. 

Author Response

Reviewer #2: The manuscript has been improved. However, there still are issues that authors must clarify. Below are the comments.

1-I am recommending major revisions because it is not clear the source of the biofilm. So, since the title, the information is not clear. In the title, the authors stated biofilm is produced by bacteria, but in the reviewer responses and in the corrected manuscript they mentioned it is obtained from a consortium of bacteria and yeast. They mentioned they used commercial Kombucha starter culture. Are the microbes in the starter culture identified? How authors can assure the biofilm is produced only by bacteria?

Response 1: The authors apologize for the confusion. Indeed, the microbes in the starter culture are not identified, the authors only know that it is a symbiotic culture of yeasts and bacteria. Nonetheless, despite the symbiotic culture, the different yeasts and bacterial species process the fermentation substrates in different and complementary manners, through competitive and cooperative interactions.

Ramírez Tapia et al. describes the Bacterial Cellulose production from the symbiotic association in kombucha as follows:

 "Moreover, it has been reported that symbiotic relations between AAB and yeasts enhance cellulose yield production due to synergic microbial metabolism. The Kombucha symbiotic community of bacteria and yeast (SCOBY) ecosystem involves cooperative and competitive interactions but overall is a symbiosis that benefits both bacteria and yeasts. Yeasts produce invertase that releases monosaccharides to media that are accessible to any microbe as a carbon source. Bacteria rapidly metabolise released sugars and occur a depletion of monomers in the environment, which in turn increases the frequency of the invertase-producing yeast. Simultaneously, bacteria produce organic acids and the surface film that protect from external competitors, by the acidification of media and the physical barrier, respectively. In addition, the ethanol produced by yeast stimulates the bacterial cellulose-synthase mechanism to produce the cellulose film (Gullo et al., 2018, May et al., 2019). Taking advantage of the dynamics of the microbial ecosystem, Kombucha tea fermentation emerges as an excellent opportunity to obtain BC due to the interplay of the SCOBY (Ramírez Tapias et al., 2020, Sharma and Bhardwaj, 2019)." (Ramírez Tapias et al., 2022).

 Ultimately, the biofilm is only produced by the bacterial genera since yeasts present in the symbiotic culture are responsible for the hydrolyzation of the carbon sources available in the fermentation medium, which are further consumed by the bacteria that synthesize the biofilm (Coelho et al., 2020; Jayabalan et al., 2014; Laavanya et al., 2021; Ramírez Tapias et al., 2022; Villarreal-Soto et al., 2018). Accordingly, Bacterial Cellulose designation is often found in the literature, even for biofilms produced from symbiotic associations of yeasts and bacteria (Domskiene et al., 2019; Leonarski, Cesca, Borges, et al., 2021; Leonarski, Cesca, Zanella, et al., 2021; Ramírez Tapias et al., 2022; Skiba et al., 2021; Soares et al., 2021).

Nonetheless, to avoid any possible confusion, the authors added the information regarding the role of yeasts and bacterial species in Bacterial Cellulose biosynthesis to Materials and Methods, section 2.2.1. Bacterial Cellulose (Page 3, line 107-111).

2-Figure 5 is not clear. There are overlapped images. 

Response 2: The authors appreciate the reviewer’s rectification, and Figure 5 has been replaced (Page 13). 

3-Figure 5A must be displayed by showing bars with the corresponding standard deviation for each treatment. Also, show the scale on Y-axis. Authors must include the results for the control treatment and carry out a means comparison test.

Response 3: The authors greatly acknowledge the reviewer’s suggestions to improve the revised manuscript. Figure 5A was changed and the datasets are now exhibited individually (Page 13). Nonetheless, since the antibacterial activity showed in the graph is expressed as Bacterial Reduction percentage, which was calculated accordingly with the equation provided in the Material and Methods section (Page 5and 6, line 228-232), Alkaline-treated BC (control) is applied in the equation to obtain the Bacterial Reduction percentage for the functionalized samples. Therefore, this dataset was not added to the graph once the bacterial reduction for the control would be 0. However, to complete the information in the revised manuscript, a table containing the CFU/mL obtained for each sample, including for the control, was added to Figure 5 (Figure 5B, page 13).

4- It could be interesting that authors provide good-quality microscopic images (if possible SEM) showing the differences in biofilm structures among the treatments. 

Response 4: The authors appreciate the reviewer’s suggestion, but unfortunately, at the moment, we can’t produce, apply the different treatments, and evaluate the biofilms through microscopic images. The evaluation provided with regard to mechanical properties showed no differences of the dyed BC in relation to un-dyed, which is also representative of no major differences in the BC structure.

Bibliography:

Coelho, R. M. D., Almeida, A. L. de, Amaral, R. Q. G. do, Mota, R. N. da, & Sousa, P. H. M. de. (2020). Kombucha: Review. In International Journal of Gastronomy and Food Science (Vol. 22, p. 100272). AZTI-Tecnalia. https://doi.org/10.1016/j.ijgfs.2020.100272

Domskiene, J., Sederaviciute, F., & Simonaityte, J. (2019). Kombucha bacterial cellulose for sustainable fashion. International Journal of Clothing Science and Technology, 31(5), 644–652. https://doi.org/10.1108/IJCST-02-2019-0010/FULL/XML

Jayabalan, R., Malbaša, R. v., Lončar, E. S., Vitas, J. S., & Sathishkumar, M. (2014). A Review on Kombucha Tea-Microbiology, Composition, Fermentation, Beneficial Effects, Toxicity, and Tea Fungus. Comprehensive Reviews in Food Science and Food Safety, 13(4), 538–550. https://doi.org/10.1111/1541-4337.12073

Laavanya, D., Shirkole, S., & Balasubramanian, P. (2021). Current challenges, applications and future perspectives of SCOBY cellulose of Kombucha fermentation. Journal of Cleaner Production, 295, 126454. https://doi.org/10.1016/J.JCLEPRO.2021.126454

Leonarski, E., Cesca, K., Borges, O. M. A., de Oliveira, D., & Poletto, P. (2021). Typical kombucha fermentation: Kinetic evaluation of beverage and morphological characterization of bacterial cellulose. Journal of Food Processing and Preservation, 45(12), e16100. https://doi.org/10.1111/JFPP.16100

Leonarski, E., Cesca, K., Zanella, E., Stambuk, B. U., de Oliveira, D., & Poletto, P. (2021). Production of kombucha-like beverage and bacterial cellulose by acerola byproduct as raw material. LWT, 135, 110075. https://doi.org/10.1016/J.LWT.2020.110075

Ramírez Tapias, Y. A., di Monte, M. V., Peltzer, M. A., & Salvay, A. G. (2022). Bacterial cellulose films production by Kombucha symbiotic community cultured on different herbal infusions. Food Chemistry, 372, 131346. https://doi.org/10.1016/J.FOODCHEM.2021.131346

Skiba, E., Gladysheva, E. K., Golubev, D. S., Budaeva, V. v., Aleshina, L., & Sakovich, G. v. (2021). Self-standardization of quality of bacterial cellulose produced by Medusomyces gisevii in nutrient media derived from Miscanthus biomass. Carbohydrate Polymers, 252, 117178. https://doi.org/10.1016/J.CARBPOL.2020.117178

Soares, M. G., de Lima, M., & Reolon Schmidt, V. C. (2021). Technological aspects of kombucha, its applications and the symbiotic culture (SCOBY), and extraction of compounds of interest: A literature review. Trends in Food Science & Technology, 110, 539–550. https://doi.org/10.1016/J.TIFS.2021.02.017

Villarreal-Soto, S. A., Beaufort, S., Bouajila, J., Souchard, J.-P., & Taillandier, P. (2018). Understanding Kombucha Tea Fermentation: A Review. Journal of Food Science, 83(3), 580–588. https://doi.org/10.1111/1750-3841.14068